# Pre-Operative Evaluation of DNA Methylation Profile in Oral Squamous Cell Carcinoma Can Predict Tumor Aggressive Potential

**DOI:** 10.3390/ijms21186691

**Published:** 2020-09-14

**Authors:** Davide B. Gissi, Viscardo P. Fabbri, Andrea Gabusi, Jacopo Lenzi, Luca Morandi, Sofia Melotti, Sofia Asioli, Achille Tarsitano, Tiziana Balbi, Claudio Marchetti, Lucio Montebugnoli

**Affiliations:** 1Section of Oral Science, Department of Biomedical and Neuromotor Sciences, University of Bologna, 40159 Bologna, Italy; davide.gissi@unibo.it (D.B.G.); andrea.gabusi3@unibo.it (A.G.); lucio.montebugnoli@unibo.it (L.M.); 2Section of Anatomic Pathology at Bellaria Hospital, Department of Biomedical and Neuromotor Sciences, University of Bologna, 40139 Bologna, Italy; viscardopaolo.fabbr2@unibo.it (V.P.F.); sofia.melotti@studio.unibo.it (S.M.); sofia.asioli3@unibo.it (S.A.); 3Section of Hygiene, Public Health and Medical Statistics, Department of Biomedical and Neuromotor Sciences, University of Bologna, 40126 Bologna, Italy; jacopo.lenzi2@unibo.it; 4Functional MR Unit, Bellaria Hospital, Department of Biomedical and Neuromotor Sciences, University of Bologna, 40139 Bologna, Italy; 5Unit of Oral and Maxillofacial Surgery, Azienda Ospedaliero-Universitaria di Bologna, Department of Biomedical and Neuromotor Sciences, University of Bologna, 40138 Bologna, Italy; achille.tarsitano2@unibo.it (A.T.); claudio.marchetti@unibo.it (C.M.); 6Unit of Anatomic Pathology, S. Orsola Hospital, 40138 Bologna, Italy; tiziana.balbi@aosp.bo.it

**Keywords:** DNA methylation, bisulfite-SEQ, oral squamous cell carcinoma, pre-operative prognostic test, oral cancer, oral oncology

## Abstract

Background: Prognosis of oral squamous cell carcinoma (OSCC) is difficult to exactly assess on pre-operative biopsies. Since OSCC DNA methylation profile has proved to be a useful pre-operative diagnostic tool, the aim of the present study was to evaluate the prognostic impact of DNA methylation profile to discriminate OSCC with high and low aggressive potential. Methods: 36 OSCC cases underwent neoplastic cells collection by gentle brushing of the lesion, before performing a pre-operative biopsy. The CpG islands methylation status of 13 gene (*ZAP70*, *ITGA4*, *KIF1A*, *PARP15*, *EPHX3*, *NTM*, *LRRTM1*, *FLI1*, *MiR193*, *LINC00599*, *MiR296*, *TERT*, *GP1BB*) was studied by bisulfite Next Generation Sequencing (NGS). A Cox proportional hazards model via likelihood-based component-wise boosting was used to evaluate the prognostic power of the CpG sites. Results: The boosting estimation identified five CpGs with prognostic significance: *EPHX3-24*, *EPHX3-26*, *ITGA4-3*, *ITGA4-4,* and *MiR193-3*. The combination of significant *CpGs* provided promising results for adverse events prediction (Brier score = 0.080, C-index = 0.802 and AUC = 0.850). *ITGA4* had a strong prognostic power in patients with early OSCC. Conclusions: These data confirm that the study of methylation profile provides new insights into the molecular mechanisms of OSCC and can allow a better OSCC prognostic stratification even before surgery.

## 1. Introduction

Oral squamous cell carcinoma (OSCC) is the tenth most common cancer in the world. The annual estimated incidence is approximately 389,000 per year [1]. Despite the improvement in surgical techniques and radio-chemotherapy, the mortality rates have remained unchanged (around 50% within 5 years of diagnosis) [2].

Prevention, early detection, and adequate pre-operative diagnosis are needed to find out patients with high-risk lesions.

OSCC has a great variety of morphological patterns that can have an impact on prognosis, as the depth of invasion, infiltrative, and growth patterns.

Pre-operative biopsy of a suspected lesion in the oral cavity is necessary to confirm the diagnosis of OSCC but can give only limited information about pathological prognostic factors. Therefore, there is a great need for strategies that can allow an accurate prognostic evaluation of OSCC aggressiveness pre-operatively.

Recently, the interest in the epigenetic mechanisms that regulate tumor development has gained increasing attention [3].

DNA methylation is a common epigenetic mechanism leading to gene silencing in tumors. Specifically, DNA methylation refers to the covalent addition of a methyl group to the 5 carbon (C5) position of cytosine bases that are located 5′ to a guanosine base in a CpG dinucleotide. CpG dinucleotides are usually found clustered in specific regions, named CpG islands, which are often located in the promoter of several genes, including tumor suppressor genes and proto oncogenes [4]. Aberrant DNA methylation in these loci may contribute to cancer progression, leading to dysregulation of mRNA expression, an early and frequent event in tumors. On the other side, DNA hypomethylation promotes tumorigenesis via transcriptional activation of oncogenes and chromosomal instability [5,6].

The loss of function of tumor-suppressor genes, which often occurs in tumors, has been ascribed more frequently to epigenetic silencing through methylation than to genetic mutations, supporting the hypothesis that epigenetic alterations have a significant role in every step of carcinogenesis [7].

Attractive strategies to better characterize OSCC cases by analyzing the methylation status of a gene panel starting from saliva and/or brushing specimens have been proposed [8,9,10,11,12,13].

Previous studies, performed by our group, demonstrated the importance of epigenetic alterations and aberrant DNA methylation of specific genes to discriminate OSCC and its precursors lesions from benign oral mucosal lesions [14,15,16]. Specifically, 19 genes known to be altered in OSCC [14,17,18,19,20] were evaluated by bisulfite Next Generation Sequencing (NGS) with the aim of developing a noninvasive procedure for OSCC detection based on oral brushing. ROC analysis of all CpGs investigated allowed us to select the highest informative ones mapped within the following 13 genes: *ZAP70*, *ITGA4*, *KIF1A*, *PARP15*, *EPHX3*, *NTM*, *LRRTM1*, *FLI1*, *MIR193*, *LINC00599*, *MIR296*, *TERT,* and *GP1BB*. A linear discriminant analysis was used to develop an algorithm of choice that clearly discriminated benign oral lesions from potentially malignant or malignant oral lesions [13,14].

None of our previous studies analyzed the impact of an altered methylation pattern of these genes obtained from pre-operative oral brushing samples on post-surgical clinical outcome. Recently, few studies from other authors demonstrated that genetic and epigenetic alterations of some genes of our 13-gene panel may play a role in OSCC prognosis, presence of metastasis and response to treatment [18,21,22,23,24,25,26]. For example, Marsit et al., starting from a case series of 68 post-surgical Formalin Fixed Paraffin Embedded (FFPE) OSCC samples revealed that an altered methylation pattern of *ZAP70* and *GP1BB* is associated with poor survival [18]. Shintani et al., starting from 7 OSCC lines, showed that an altered methylation pattern of *FLI1* is a prediction marker gene for OSCC radiotherapy resistance [22].

The aim of the present study is to correlate the same epigenetic alterations with prognosis. To this purpose, samples were collected before surgery through oral brushing in a group of OSCC patients. The relationship between methylation profile of each gene, clinic-pathological features and follow-up was evaluated. Most informative CpGs were also used to generate a unique prognostic model in OSCC patients based on a Cox proportional hazards model via likelihood-based component-wise boosting.

## 2. Results

Thirty-six patients met the inclusion criteria. Median follow-up time was 25.8 months (2.1 years), ranging from 2.8–63.1 months (0.2–5.3 years). In total, one third of the patients (*n* = 12) experienced disease relapse during follow-up. The Kaplan–Meier estimate of recurrence-free survival for the entire study sample is illustrated in Figure 1.

### 2.1. Pathological Findings and Tumor Stage 

Evaluation of TNM was assigned according to p-TNM Classification of Tumors (AJCC 8th Edition) [27]. Tumor size (T): 17 patients were T1 (47%), 10 patients were T2 (28%), 3 patients were T3 (8%), 6 patients were T4 (17%). Lymph nodes involvement (N): 32 patients were N0 (89%); N1: 2 patients (6%); N2: 1 patient (3%); N3: 1 patient (3%).

Sixteen patients were diagnosed as T1 N0, 1 as T1 N2, 10 as T2 N0, 1 as T3N0, 2 as T3 N1, 5 as T4 N0 and 1 as T4 N3 (see Table 1 for details).

The log-rank test revealed that the only clinic-pathological factor significantly associated with worse prognosis was higher pattern of invasion (P3-P4 according to Chang et al. classification [28]). 

### 2.2. Methylation Profile and Clinical-Pathological Characteristics

Tumor size: the Mann–Whitney *U* test showed higher methylation levels of most CpG sites of *LINC00599* (coordinates hg38: Chr8: 9903242-9903378) in T1-T2 tumors as compared to T3-T4 tumors, although results failed to achieve statistical significance at the 5%-level (Figure 2).

Lymph-node metastases: As shown in Figure 3, a higher methylation profile of CpG islands in *EPHX3* was found in OSCC without lymph-node involvement as compared to OSCC with a positive lymph-node status at the diagnosis (Figure 3), although results failed to achieve statistical significance.

The Mann–Whitney U test and Kruskal–Wallis test did not show other differences in the methylation profile of 13 genes in relationship with other clinical-pathological characteristics of the studied population. 

### 2.3. Methylation Profile and Adverse Event (AE)

As possible molecular predictors, we analyzed the methylation data of 238 CpG sites from 13 genes (*ZAP70*, *ITGA4*, *KIF1A*, *PARP15*, *EPHX3*, *NTM*, *LRRTM1*, *FLI1*, *MiR193*, *LINC00599 TERT*, *MiR296,* and *GP1BB*). Permutation-based *p*-value estimates for each candidate covariate are presented in Appendix A. The boosting estimation resulted in five non-zero regression coefficients associated with *EPHX3-24*, *EPHX3-26*, *ITGA4-3*, *ITGA4-4,* and *MiR193-3*. As shown in Table 2, *EPHX3* was associated with worse prognostic outcomes in case of hypomethylation (HR < 1), while *ITGA4* and *MiR193* were associated with worse prognosis outcomes in case of hypermethylation (HR > 1). The coefficient estimates (ln HR) at each boosting step are illustrated in Appendix A. The integrated Brier score of the final risk prediction model was 0.080, the *C*-index was 0.802 and the integrated AUC was 0.850 (Table 2, Figure 4).

Removing the connection information from the boosting algorithm, results were quite stable (Appendix A): the CpG sites retained in the final Cox model were *EPHX3-23*, *EPHX3-24*, *ITGA4-4* and *MiR193-3*; the discriminative ability was virtually identical (*C*-index = 0.805; integrated AUC = 0.847), as well as the overall performance (integrated Brier score = 0.089). As shown in Appendix A and Appendix A, re-running all analyses after converting Beta-values into *M*-values, the overall performance of the regression model was similar, but its calibration and discriminative ability were poorer in the last months and in the first months of follow-up, respectively.

In the subgroup analysis of patients with early tumor diagnosis (26 patients with T1-2N0), we found that *ITGA4* had a strong prognostic power for future disease recurrence (Table 3).

## 3. Discussion

Maxillo-facial surgeons and oncologists are called not only to surgically treat an OSCC but also to prevent AEs that could affect their patients. Therefore, pre-operative assessment plays a crucial role in OSCC work-up. 

Clinical evaluation and radiological assessment (CT-scan and MRI) are essential for tumor staging. 

A pre-operative biopsy is required to confirm the diagnosis of OSCC and to give information about pathological prognostic factors. It is now clear that clinic-pathological findings, as well as radiological evaluation, alone are not sufficient in predicting OSCC clinical behavior. Local recurrences, nodal metastases, or second primary tumors can hamper the prognosis particularly in early staged OSCC. If the poor prognosis for locally advanced tumors is well explainable, also T1-T2 stages can be occasionally unpredictable in terms of disease control [29]. Despite an apparent favorable condition, sometimes early staged OSCC can have a very aggressive biological behavior, even if resected with free surgical margins.

As in other body sites (as lung or colon), OSCC molecular profile can give interesting information about tumor aggressiveness: Specifically, gene silencing by promoter methylation seems to play a crucial role in determining tumor aggressive potential [30].

In this study, the relationship between the methylation profile of a panel of 13 genes (*ZAP70*, *ITGA4*, *KIF1A*, *PARP15*, *EPHX3*, *NTM*, *LRRTM1*, *FLI1*, *MIR193*, *LINC00599 TERT*, *MIR296*, and *GP1BB*) and their role in predicting some factors (tumor size, lymph node involvement, AE), not otherwise identifiable from pathological pre-operative diagnosis was investigated. The same gene panel was recently validated as diagnostic marker of OSCC and proved to be accurate in the early detection of oral cancer [15].

The use of bisulfite NGS method allowed to test many CpG sites for a single gene and to align longer reads to the reference sequence more easily and with greater accuracy. Superiority of amplicon bisulfite sequencing in terms of accuracy, robustness, and throughput compared to other DNA methylation analysis methods such as methylation-sensitive PCR was recently reported by Bock et al. [31]. Furthermore, DNA methylation analysis was assessed starting from minimal invasive collection specimens obtained by oral brushing. This approach may be advantageous for its very limited invasiveness and because it avoids formalin fixation of the neoplastic cells. It is well known that DNA extraction and consequent DNA methylation analysis can be easier performed in fresh-unfixed material. Furthermore, brushing cell collection from the whole tumor area can provide a sample representative of the complex epigenetic landscape of each single OSCC lesion. In this setting intratumoral heterogeneity can be evaluated pre-operatively.

The comparative analysis of methylation profiles with clinical-pathological characteristics suggested the presence of an association, albeit not significant at the 5%-level, between *LINC00599* and T stage. In particular, several CpG islands in exon 1 of *LINC00599* segregated between early-OSCC (T1–2) and locally advanced-OSCC (T3–4). *LINC00599* (Long Intergenic Non-Protein Coding RNA 599) is a long noncoding RNA Gene mapped close to the *MIR124* gene. A number of studies have reported that *LINC00599* exerts important regulatory functions in cell proliferation differentiation, and in atherosclerosis [32,33]. Little is known about the role of *LINC0059* methylation in oral cancer. This finding supports the idea that changes in epigenetic landscape occur during tumor progression. However, further studies and larger studies are needed to elucidate contribution *LINC00599* methylation in oral carcinogenesis. Similarly, there was a tendency toward hypomethylation for most CpG islands of *EPHX3* in case of positive lymph-node status at the diagnosis.

In relation to post-surgical clinical outcomes, the analysis of methylation profiles identified 19/238 CpG sites significantly associated with the occurrence of loco regional adverse events (see Appendix A). It also emerged that only 4 out of 13 studied genes in the panel harbored CpGs with prognostic relevance (9 CpG islands for *EPHX3*, 7 CpG islands for *ITGA4*, and only one CpG island respectively for *MiR193*, *MiR296,* and *TERT*). The boosting estimation resulted in five non-zero regression coefficients associated with *EPHX3-24*, *EPHX3-26*, *ITGA4-3*, *ITGA4-4,* and *MiR193-3* significantly associated with the occurrence of loco regional adverse events. Boosting is a method for incrementally building linear combinations of “weak” models, to generate a “strong” predictive model. This process shows similarities to lasso-like approaches, with many of the estimated coefficients shrunk toward zero. Boosting algorithms have been shown to be particularly useful to handle models in which the number of candidate predictors exceeds the number of observations (high-dimensional settings).

Putative prognostic predictors methylation analysis showed that *EPHX3* was associated with worse prognostic outcomes when hypomethylated (HR < 1), while *ITGA4* and *MiR193* were associated with worse prognosis outcomes in case of hypermethylation (HR > 1).

Prior studies have noted that the methylation of epoxide hydrolase 3 (*EPHX3*) is significantly associated with the prognosis of prostate cancer and its inclusion into clinical practice has been suggested as a tool to a more accurate prediction of which patients may experience prostate cancer recurrence. Curiously, it was found hypermethylated in human salivary gland Adenoid Cystic Carcinoma [34]. In OSCC its role is yet to be elucidated but emerging evidence from previous published research papers and the relationship with lymph-node involvement and occurrence of secondary neoplastic events suggest its promising use in risk predictive models. 

*ITGA4* is involved in gastric and colorectal cancer [35,36]. Previous reports indicate *ITGA4* as a promising marker for oral squamous cell carcinoma and especially tongue oral cancer. 

*MiR193* was previously identified as associated with oral cancers in previous studies by Kozaki and our group [15,19].

These findings suggest that patients experiencing less favorable behavior of the disease may actually have a distinctive methylation profile. Noteworthy, *ITGA4* showed a strong prognostic power for future disease recurrence in the subgroup analysis of patients with early stage tumor diagnosis. T1-T2 are commonly considered at lower risk of unfavorable outcome but they can pose a problem in the correct planning of surgical management. Despite the apparent ease to manage these early staged tumors, sometimes the surgeon faces to unpleasant surprises during the follow-up period. In fact, local recurrence can be experienced even if resection margins were clear and neck nodal metastases can appear in 10–30% of clinically negative necks (cN0) patients. Indeed, neck management for cT1-2 N0 patients remains debated. In these cases, known prognostic clinical-pathological factors (grading, depth of invasion, surgical margins, pattern of invasion) are usually less informative as compared to advanced tumors. The presence of a biomarker with the ability of recognition of early tumors that will eventually show aggressive behaviors before surgery starting from pre-operative oral brushing specimen is of primary importance in oral cancer management.

Nevertheless, the great amount of data deriving from several CpGs makes methylation analysis hardly applicable for models of single patient risk assessment. For this reason, in the last decade researchers have called in biostatistical techniques to translate complex results into clinical practice.

In the present study, the combination of significant CpGs (*EPHX3-24*, *EPHX3-26*, *ITGA4-3*, *ITGA4-4,* and *MiR193-3)* to build up a risk assessment prediction model provided promising results as shown by its performance and predictive accuracy. Interestingly, when connection information from the boosting algorithm was removed, *EPHX3-24*, *ITGA4-4,* and *MiR193-3* retained a strong predictive prognostic ability in the final Cox model. More in detail, *EPHX3* gene showed fluctuation and changes in significant CpGs depending on which model was used (*EPHX3-18* to *EPHX3-26).* However, prognostic ability of CpGs in the area resulted virtually equivalent. In addition, the conversion of Beta-values into M-values resulted in similar but poorer overall performance of the regression model, suggesting the use of Beta values.

More in detail, *EPHX3* gene showed fluctuation and changes in significant CpGs depending on which model was used (*EPHX3-18* to *EPHX3-26).* However, prognostic ability of CpGs in the area resulted almost equivalent.

In addition, the conversion of Beta-values into M-values, resulted in similar but poorer overall performance and discriminative ability of the regression model suggesting a preferable use of Beta values.

Combination of epigenetic profiles of several genes as useful source of information is in agreement with emerging published literature in which several authors have tried to identify most relevant methylated genes to be used for prognostic purposes. Shen et al. in 2017 identified an OSCC DNA-methylation signature overall survival, based on 7 significant CpGs from genes *AJAP1*, *SHANK2*, *FOXA2*, *MT1A*, *ZNF570*, *HOXC4,* and *HOXB4* [37].

Zhu in 2019 found out that the combined analysis of the methylation status of 5 genes (*CENPV*, *SYTL2*, *OCLN*, *CASD1,* and *TUB*) helped in the construction of a risk prediction model of the prognosis of OSCC [38] 

Pan et al. in 2019 reported that subsequent combined survival analysis on six genes (*INA*, *LINC01354*, *TSPYL4*, *MAGEB2*, *EPHX3*, and *ZNF134*) could be used as independent prognostic markers and potentially used as drug targets [39].

A limit of the present study is the low number of patients, especially the number of advanced OSCC tumors (T3-T4). However, component-wise boosting was specifically used to overcome the limits imposed by the small sample size. A second limit of the present study is the absence of a prospective test set of patients that we hope to undertake in future investigations in order to provide an unbiased predictive evaluation of the fitted model. A validated Cox model including molecular and clinical predictors has the potential to estimate for each patient the risk of disease relapse at any given time points following surgery.

## 4. Materials and Methods 

### 4.1. Ethics Statement

All clinical investigations were conducted according to the principles of the Declaration of Helsinki. The study was approved by the local ethics committee (520/2018/Sper/AOUBo, 2018/Dec/12th). All information regarding the human material used in this study was managed using anonymous numerical codes.

### 4.2. Study Setting and Data Collection

Selection criteria were the following: oral brushing performed at the time of pre-operative biopsy; histological diagnosis of primary oral squamous cell carcinoma; no other neoplastic diseases reported on anamnesis; radical surgery with histologically negative resection margins; availability of follow-up information.

All these cases were treated at the Departments of Oral Sciences and Oral and Maxillofacial Surgery, University of Bologna, from March 2013 to December 2018. Histological diagnoses were performed at the Section of Anatomic Pathology at Bellaria Hospital, Department of Biomedical and Neuromotor Sciences, University of Bologna (VPF, SA) and Unit of Anatomic Pathology, S. Orsola Hospital, Bologna (TB).

All patients presenting with a suspected oral neoplastic lesion underwent oral brushing sampling before diagnostic incisional biopsy. Oral brushing was performed according to a previously described protocol [14,15].

After surgery, histological examination was performed blindly from the methylation profile. A multi-head microscope discussion was made on discordant cases to obtain a common diagnosis. Histological diagnoses and staging were established following currently accepted criteria [27,40].

### 4.3. Treatment Modality

After the diagnostic workup and multidisciplinary discussion, all 36 patients underwent surgical resection of OSCC in accordance with standard treatment practice [41]. Surgery consisted of composite resections, including excision of the primary tumor with ipsilateral or bilateral neck dissection, in accordance to the N-status. Both primary closure and local flaps were performed for early stages. Microvascular reconstruction was performed for patients with locally advanced disease. Post-operative radiation therapy was performed according to currently accepted criteria [42].

Follow-up was performed every two weeks for the first two months after surgery and then monthly during the first year after surgery, every three months during the second year after surgery, and finally every six months. A CT scan or MRI was requested every six months during the first three years after surgery and then once a year.

The disease-free survival endpoints, defined as the duration between oral brushing cell collection and the diagnosis of local recurrence, lymph node or distant metastasis, death, or the last follow-up visit, were evaluated in June 2020.

### 4.4. DNA Methylation Analysis

DNA methylation analysis was performed as previously described by Morandi et al. [15]. Shortly, DNA from exfoliating brush specimens was purified using The MasterPure™ Complete DNA Purification Kit (Lucigen, Middleton, WI, USA, cod. MC85200), and treated with sodium bisulfite using the EZDNA Methylation-Lightning™ Kit (ZymoResearch, Irvine, CA, USA, cod. D5031) according to the manufacturer’s instructions. Quantitative DNA methylation analysis was performed by next-generation sequencing for the following genes: *ZAP70*, *ITGA4*, *KIF1A*, *PARP15*, *EPHX3*, *NTM*, *LRRTM1*, *FLI1*, *MIR193*, *LINC00599*, *MIR296*, *TERT,* and *GP1BB* (Table 4).

Libraries were prepared using the Nextera™ Index Kit (Illumina, San Diego, CA, USA, FC-121-1012) following a two steps approach with locus-specific bisulfite amplicon [15] and loaded onto MiSEQ (Illumina, San Diego, CA, USA, cod. 15027617). Each NGS experiment was designed to allocate at least 1000 reads/amplicon, with the aim to reach a depth of coverage of 1000×. FASTQ output files were processed for quality control (>Q30) and converted into FASTA format in a Galaxy Project environment [43]. The methylation ratio of each CpG was calculated in parallel by different tools: BSPAT (http://cbc.case.edu/BSPAT/index.jsp) [44] BWAmeth in a Galaxy Project environment (Europe) followed by the Methyl Dackel tool (https://github.com/dpryan79/MethylDackel), EPIC-TABSAT [45], and finally Kismeth [46]. Methylation plotter tool was used to compare DNA methylation level of 13 gene panel and clinic-pathological variables [47].

### 4.5. Statistical Analysis

Summary statistics for the study sample were presented as frequencies and percentages. Recurrence-free survival was calculated using the date of surgery as the time origin. The endpoints that we considered were local recurrence, second primary tumor, lymph-node metastasis, and disease-related death, whichever occurred first. Lost-to-follow-up were right-censored at the time of the status last known. The survival function was estimated with the Kaplan–Meier method.

Differences in methylation levels across clinical-pathological groups were evaluated with the Mann–Whitney *U* test. Permutation-based *p*-values were corrected using Simes’ method for false discovery rate control.

To evaluate the prognostic power of the CpG sites, we fit a Cox proportional hazards model via likelihood-based component-wise boosting [48,49]. This forward procedure, which is rooted in the field of machine learning [50], starts with an empty model and updates only one regression coefficient in each step of the algorithm. Updates are done by adding small fractions of the estimated regression coefficients to the estimates obtained in previous steps; these fractions are governed by a penalty term attached to the log-likelihood (usually 0.1). This process results in sparse fits similar to lasso-like approaches, with many of the estimated coefficients shrunk toward zero. Boosting algorithms have been shown to be particularly useful to handle models in which the number of candidate predictors exceeds the number of observations (high-dimensional settings) [51,52].

The number of boosting steps, which is the main tuning parameter of the procedure and determines both shrinkage of effect estimates and variable selection, was optimized by cross-validation [52]. To deal with the high variability of the results due to small sample sizes, we averaged over 100 repetitions of cross-validation, then repeated the same procedure by changing the number of partitions (leave-one-out, 3-, 6-, and 9-fold) and selected the median value among the four penalty parameters [53]. This approach led to an optimal boosting step number of seven. All methylation Beta-values specified in the Cox proportional hazards model were standardized in order to meet the homoscedasticity requirement of the likelihood-based boosting algorithm [52]. Permutation-based *p*-value estimates were calculated for each candidate CpG site [54]. By including pathway information in boosting estimation, CpG sites were more likely to be chosen in subsequent boosting steps if a connected CpG site from the same gene had been chosen in an earlier boosting step [55].

The overall performance of the risk prediction model resulting from seven boosting steps was evaluated with the Brier score, which is a weighted average of the squared distances between the observed survival status and the survival probability predicted by the model. A prediction error curve was obtained by following the Brier score over time, and the integral under the curve was calculated as a summary measure for model performance; the lower the curve, the better is the performance of the model [56,57]. The discriminative ability of the model was evaluated with the C-index for survival data, a measure that quantifies the proportion of all patient pairs for whom the predicted and observed survival outcomes are concordant. A C-index = 0.5 indicates a non-informative prediction rule, while a C-index = 1 indicates perfect association. We used Uno’s estimate of the index, which is based on inverse-probability-of-censoring weights and has been shown to be more robust than Harrell’s estimate to the presence of many censored observations [58]. We also estimated the time-dependent area under the receiver operator characteristic (ROC) curve (AUC) as an additional measure of discriminative ability [59], because it is more robust than the C-index to deviations from the proportional hazards assumptions [60]. The summary measure for time-dependent AUC was the integrated AUC, which averages all available statistics over time. It should be recognized that the performance of the risk prediction model was evaluated using the original training set of 36 patients, therefore all our estimates have to be considered as overly optimistic.

In a sensitivity analysis, we removed connection information from the algorithm in order to treat all CpG sites as unconnected covariates. In another sensitivity analysis, Beta-values were converted into M-values, i.e., the binary logarithm of the intensities of methylated probes versus non-methylated probes [61]. In a subgroup analysis, the prognostic power of the CpG sites was reevaluated on the 26 study patients with an early diagnosis of OSCC. All analyses were conducted using R version 4.0.1 [62]; likelihood-based boosting was performed with the R package CoxBoost [63].

## 5. Conclusions

The present preliminary study, even if performed in a small population of patients affected by Oral Cancer, confirms the attractive use of the preoperative evaluation of methylation profile also for the prognostic assessment of patients with OSCC. Adding more details than the simple biopsy, molecular findings from oral brushing could help clinicians to stratify patients at high- versus low-risk of recurrences, metastases and second tumors, and to plan the adequate treatment. Further studies with a larger and homogeneous cohort are needed to elucidate the intrinsic prognostic potential of our assay.

## Figures and Tables

**Figure 1 ijms-21-06691-f001:**
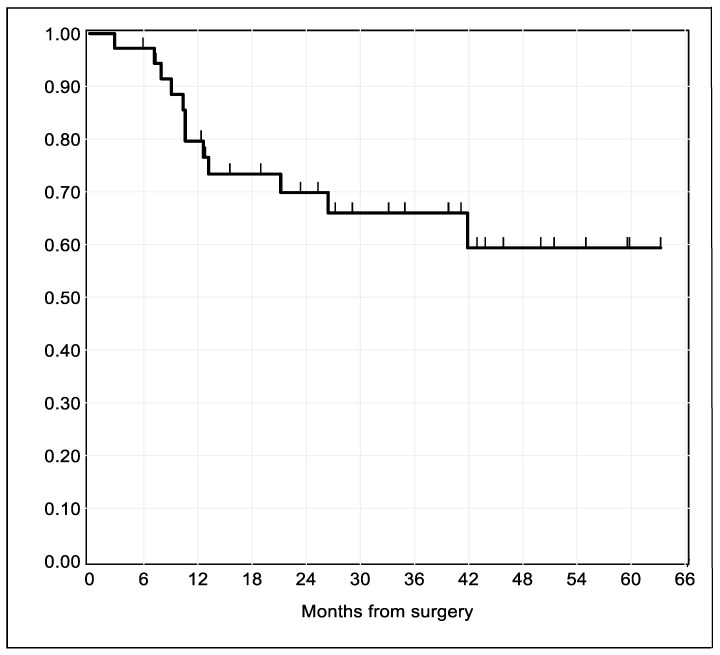
Kaplan–Meier survival estimates of time to relapse after surgical resection for oral squamous cell carcinoma (OSCC). The spikes indicate censoring times.

**Figure 2 ijms-21-06691-f002:**
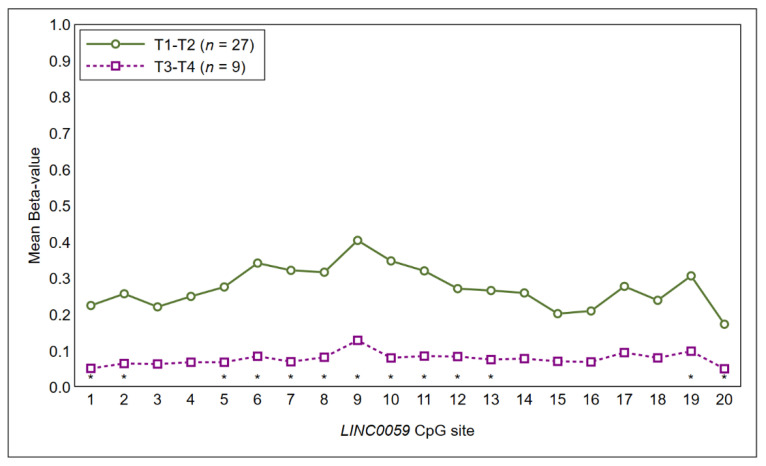
Methylation profile of the CpG sites of LINC00599 segregated between T1-T2 tumors and T3-T4 tumors. Statistical significance at 10% is marked with an asterisk.

**Figure 3 ijms-21-06691-f003:**
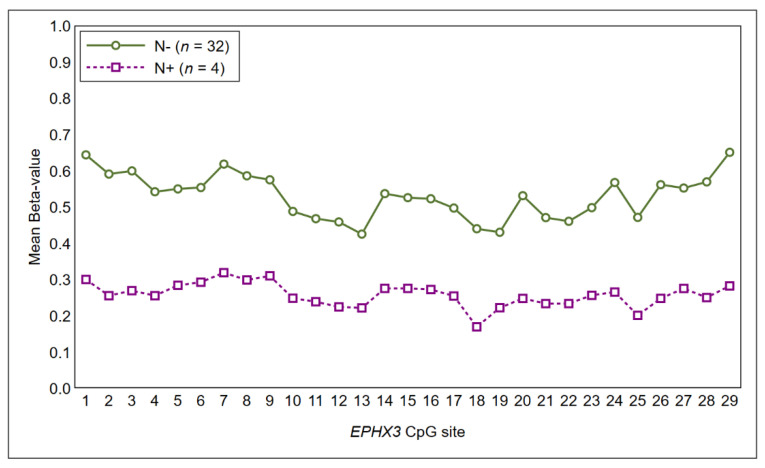
Methylation profile of the CpG sites of *EPHX3* segregated between N− tumors and N+ tumors.

**Figure 4 ijms-21-06691-f004:**
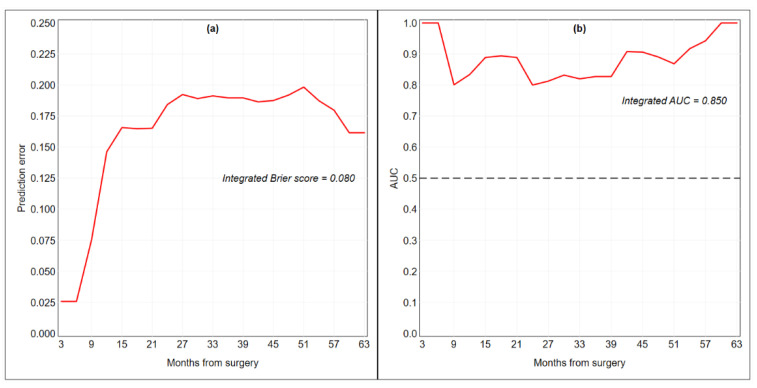
Time-dependent Brier score (a) and area under the receiver operator characteristic curve (AUC) (b) for the final Cox proportional hazards model fitted by component-wise likelihood-based boosting. The dashed line represents a non-discriminatory model (random guessing). Performance and predictive accuracy of the model were estimated using the original training set of 36 patients.

**Table 1 ijms-21-06691-t001:** Clinical-pathological features of the studied population: Asterisks (*) indicate significant prognostic variables of loco-regional secondary neoplastic manifestation at the 5%-level. *p*-values were obtained with the log-rank test.

Clinical-Pathological Variables				
		Patients	Relapses Observed	*p*-Value
Sex	Male	17 (47%)	6 (35%)	0.82
	Female	19 (53%)	6 (31%)	
Age	<65	15 (42%)	8 (53%)	0.35
	>65	21 (58%)	4 (19%)	
Smoke	Yes	7 (19%)	3 (43%)	0.26
	No	29 (81%)	9 (31%)	
Site	Tongue and floor of mouth	13 (36%)	4 (31%)	0.96
	Buccal and labial mucosa	7 (19%)	3 (43%)	
	Gingiva, Hard Palate, Retromolar region	16 (45%)	5 (31%)	
T stage	T1-T2	27 (75%)	8 (29%)	0.33
	T3-T4	9 (25%)	4 (44%)	
N stage	N−	32 (89%)	9 (28%)	0.06
	N+	4 (11%)	3 (75%)	
Grading	G1	20 (56%)	5 (25%)	0.18
	G2	14 (39%)	6 (43%)	
	G3	2 (5%)	1(50%)	
Surgical margins	Free	29 (81%)	9 (31%)	0.83
	Close	4 (11%)	1 (25%)	
	Displasia	3 (8%)	2 (66%)	
	Involved	0 (0%)		
Presence of associated OPMD	None	26 (72%)	11 (42%)	0.07
	Lichen	6 (17%)	0 (0%)	
	Leucoplakia	4 (11%)	1 (25%)	
Depth of invasion (DOI)	<4 mm	21 (58%)	5 (24%)	0.07
	>4 mm	15 (42%)	7 (47%)	
Pattern of invasion	P1-P2	22 (61%)	4 (18%)	0.01 *
	P3-P4	14 (39%)	8 (57%)	
Radiotherapy	Yes	8 (22%)	4 (50%)	0.2
	No	28 (78%)	8 (29%)	

**Table 2 ijms-21-06691-t002:** Cox proportional hazards model resulting from likelihood-based component-wise boosting after seven steps. The five non-zero regression coefficients (ln HR) are presented along with hazard ratios (HRs) and permutation-based *p*-values. Coefficients are scaled to be at the level of the original methylation Beta-values.

CpG Site	Ln HR	HR	*p*-Value
*EPHX3-24 (Chr19:15232040)*	−0.0234	0.9769	0.0157
*EPHX3-26 (Chr19:15232034)*	−0.0226	0.9777	0.0172
*ITGA4-3 (Chr2:181458175)*	0.0163	1.0165	0.0078
*ITGA4-4 (Chr2:181458181)*	0.0306	1.0310	0.0027
*MiR193-3 (Chr17:31559856)*	0.0089	1.0090	0.0099
Integrated Brier score	0.080		
C-index	0.802		
Integrated AUC	0.850		

**Table 3 ijms-21-06691-t003:** Cox proportional hazards model resulting from likelihood-based component-wise boosting after nine steps. The study sample is made of patients with an early diagnosis of OSCC (*n* = 26). The four non-zero regression coefficients (ln HR) are presented along with hazard ratios (HRs) and permutation-based *p*-values. Coefficients are scaled to be at the level of the original methylation Beta-values.

CpG Site	Ln HR	HR	*p*-Value
*ITGA4-3(Chr2:181458175)*	0.0616	1.0636	0.0001
*ITGA4-4(Chr2:181458181)*	0.0467	1.0478	0.0003
*ITGA4-7(Chr2: 181458229)*	0.0176	1.0177	0.0046
*ITGA4-12(Chr2: 181458289)*	0.0152	1.0153	0.0052
Integrated Brier score	0.059		
C-index	0.892		
Integrated AUC	0.903		

**Table 4 ijms-21-06691-t004:** List of 13 genes target.

*Gene*	Map	Position	Amplicon Length	Position Respect to TSS	Number of Interrogated CpG	hg38 Coordinates
***ZAP70***	2q11.2	Exon 3	180	10728	20	Chr2: 97724265-97724445
***GP1BB***	22q11.21	Exon 1	192	363	18	Chr22: 19723282-19723460
***KIF1A***	2q37.3	Exon 1	189	−1	27	Chr2: 240820168-240820310
***PARP15***	3q21.1	Exon 1	206	93	19	Chr3:122577695-122577901
***ITGA4***	2q31.3	Exon 2	214	912	14	Chr2:181457647-181457879
***NTM***	11q25	Exon 1	190	62	15	Chr11:131911126-131911314
***MIR193A***	17q11.2	Promoter	256	−178	26	Chr17:31559818-31560073
***EPHX3***	19p13.12	Exon 1	223	215	29	Chr19:15231995-15232217
***LINC00599***	8p23.1	Exon 1	199	69	20	Chr8:9903205-9903403
***FLI1***	11q24.3	Exon 1	186	187	12	Chr11:128694103-128694288
***MIR296***	20q13.32	Exon 1	238	180	15	Chr20:58817149-58817363
***LRRTM1***	2p12	Promoter	179	−431	24	Chr2:80304527-80304705
***TERT***	5p15.33	Intron4-5	109	14976	6	Chr5:1279604-1279759

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
