# Peer review of "Pre-Operative Evaluation of DNA Methylation Profile in Oral Squamous Cell Carcinoma Can Predict Tumor Aggressive Potential"

_ijms, 2020, doi:10.3390/ijms21186691_

Round 1

Reviewer 1 Report

This study describes the potential use of methylation markers to predict prognosis. A panel of 13 markers was tested in a small group of patients.

Although genetic markers surely play a role in tumor behaviour, it is important to realise that to prove any involvement, either there should be a hypothesis to test (involvement is some important pathway), or the series should be large and as homogenous as possible. Also, these factors should be proven to be independent of known prognostic factors, like tumor stage.

The current markers have shown theirs value in being reliable markers for the presence of tumor. It would be ideal if the same markers also predict outcome and guide treatment.

Unfortunately, this series is too small to really make any conclusion on a prognostic significance of these markers.

Author Response

Dear Editor and Reviewers,

We thank you for your interest in our research and for the time you spent reviewing it. We have carefully considered the comments provided and have revised the paper accordingly. All changes are highlighted in red in the manuscript.

Reviewer 1

This study describes the potential use of methylation markers to predict prognosis. A panel of 13 markers was tested in a small group of patients.

 Although genetic markers surely play a role in tumor behavior, it is important to realize that to prove any involvement, either there should be a hypothesis to test (involvement is some important pathway), or the series should be large and as homogenous as possible. Also, these factors should be proven to be independent of known prognostic factors, like tumor stage.

The current markers have shown theirs value in being reliable markers for the presence of tumor. It would be ideal if the same markers also predict outcome and guide treatment.

Unfortunately, this series is too small to really make any conclusion on a prognostic significance of these markers.

Answers to the Reviewer 1:

The rationale of the present study was that:

  1. Previous studies, performed by our group, clearly demonstrated that methylation profiles of ZAP70, ITGA4, KIF1A, PARP15, EPHX3, NTM, LRRTM1, FLI1, miR193, LINC00599, miR296, TERT, and GP1BB analyzed by bisulfite Next Generation Sequencing (NGS) from oral brushing samples discriminated malignant or premalignant oral lesions from healthy donors.
  2. None of our previous studies analyzed the impact of an altered methylation pattern of these genes obtained from pre-operative oral brushing samples on post-surgical clinical outcome. Recently, few studies from other authors demonstrated that genetic and epigenetic alterations of some genes out of our 13-gene panel may play a role in OSCC prognosis, presence of metastasis and response to treatment [17–23]. For example, Marsit et al., starting from a case series of 68 post-surgical Formalin Fixed Paraffin Embedded (FFPE) OSCC samples revealed for the first time that an altered methylation pattern of ZAP70 and is associated with poor survival [17]. Shintani et al., starting from 7 OSCC lines, showed that an altered methylation pattern of FLI1 is a prediction marker gene for OSCC radiotherapy resistance [19].
  3. The aim of the present study is to correlate the same epigenetic alterations with prognosis. To this purpose, samples were collected before surgery through oral brushing in a group of OSCC patients. The relationship between methylation profile of each gene, clinical-pathological features and follow-up was evaluated.

The introduction section at page 2 from line 73 to 88 has been modified to better clarify the rationale of the present study.

The reviewer also sustains that this population study is too small to really make any conclusion on a prognostic significance and it should be large and as homogeneous as possible. We perfectly agree that a larger population study is necessary to confirm our preliminary data or to really make any definitive conclusion. However, in the present study to overcome the limits of the small population study, we decided to fit a Cox proportional hazards model via likelihood-based component-wise boosting, as reported in the statistical analysis section (page 11 line 354-371) and in the discussion section (page 7 lines 218-233). Boosting is a method for incrementally building linear combinations of “weak” models, to generate a “strong” predictive model and boosting algorithms have been shown to be particularly useful to handle models in which the number of candidate predictors exceeds the number of observations (high-dimensional settings), similarly to our study.  

However, in the discussion section at page 9 line 290 and in the conclusion at page 12 line 408 we modified the following sentence in order to underline the limits of the present study related to population study.

“A limit of the present study is the low number of patients, especially the number of advanced OSCC tumors (T3-T4). However, component-wise boosting was specifically used to overcome the limits imposed by the small sample size. A second limit of the present study is the absence of a prospective test set of patients that we hope to undertake in future investigations in order to provide an unbiased predictive evaluation of the fitted model. A validated Cox model including molecular and clinical predictors has the potential to estimate for each patient the risk of disease relapse at any given time points following surgery.”

“The present preliminary study, even if performed in a small population of patients affected by Oral Cancer, confirms the attractive use of the preoperative evaluation of methylation profile also for the prognostic assessment of patients with OSCC. Adding more details than the simple biopsy, molecular findings from oral brushing could help clinicians to stratify patients at high- versus low-risk of recurrences, metastases and second tumors, and to plan the adequate treatment. Further studies with a larger and homogeneous cohort are needed to elucidate the intrinsic prognostic potential of our assay.”

The reviewer finally sustains that epigenetic markers should be proven to be independent of known prognostic factors, like tumor stage. Nevertheless, in the present small population study the only clinical-pathological variable significantly related with prognosis resulted the presence of a higher pattern of invasion that can be evaluated only after demolitive surgery through the microscopic analysis of surgical specimen. The altered methylation status of 5 CpG islands (EPHX3-24, EPHX3-26, ITGA4-3, ITGA4-4 and MiR193-3) resulted the only pre-operative clinic-pathological variable significantly related with worse prognosis. In Table 3 we reported data on Kaplan Meier analysis to evaluate the prognostic role of clinical-pathological factors.

Reviewer 2 Report

In this study, authors evaluated the prognostic impact of DNA methylation profile to discriminate oral squamous cell carcinoma (OSCC) with high and low aggressive potential. Authors examined the CpG islands methylation status of 13 gene (ZAP70, ITGA4, KIF1A, PARP15, EPHX3, NTM, LRRTM1, FLI1, MiR193, LINC00599, MiR296, TERT, GP1BB) by bisulfite Next Generation Sequencing (NGS) in 36 OSCC cases. Authors identified five CpGs (EPHX3-24, EPHX3-26, ITGA4-3, ITGA4-4 and MiR193-3) with prognostic significance. Moreover, the combination of significant CpGs can be used for adverse events prediction. In addition, ITGA4 was a strong prognostic factor in patients with early OSCC. Overall, authors suggest that methylation profile provides new insights into the molecular mechanisms of OSCC and can allow a better OSCC prognostic stratification.

This paper contains interesting findings. However, authors should answer the following comments. If authors satisfactory respond to these comments, this work is suitable for publication.

My comments are the following:

  1. In this study, authors examined the CpG islands methylation status of a 13 genes set (ZAP70, ITGA4, KIF1A, PARP15, EPHX3, NTM, LRRTM1, FLI1, MIR193, LINC00599, MIR296, TERT, and GP1BB). Authors described that “previous studies, performed by our group, demonstrated the importance of epigenetic alterations and aberrant DNA methylation of specific genes to discriminate OSCC and its precursors lesions from benign oral mucosal lesions”. Please explain the detail of why authors picked up 13 genes.
  2. Authors examined the CpG islands methylation status in 36 OSCC cases. However, 75% (27 cases) are early stage (T1-T2). It is better to increase the number of advance cases (T3 & T4).
  3. Authors examined the methylation profile and clinical-pathological characteristics including tumor size and LN metastasis. How about other clinical-pathological characteristics? Ex) differentiation, invasion pattern, and/or recurrence…

Author Response

Dear Editor and Reviewers,

We thank you for your interest in our research and for the time you spent reviewing it. We have carefully considered the comments provided and have revised the paper accordingly. All changes are highlighted in red in the manuscript.

Reviewer 2:

In this study, authors evaluated the prognostic impact of DNA methylation profile to discriminate oral squamous cell carcinoma (OSCC) with high and low aggressive potential. Authors examined the CpG islands methylation status of 13 gene (ZAP70, ITGA4, KIF1A, PARP15, EPHX3, NTM, LRRTM1, FLI1, MiR193, LINC00599, MiR296, TERT, GP1BB) by bisulfite Next Generation Sequencing (NGS) in 36 OSCC cases. Authors identified five CpGs (EPHX3-24, EPHX3-26, ITGA4-3, ITGA4-4 and MiR193-3) with prognostic significance. Moreover, the combination of significant CpGs can be used for adverse events prediction. In addition, ITGA4 was a strong prognostic factor in patients with early OSCC. Overall, authors suggest that methylation profile provides new insights into the molecular mechanisms of OSCC and can allow a better OSCC prognostic stratification.

 This paper contains interesting findings. However, authors should answer the following comments. If authors satisfactory respond to these comments, this work is suitable for publication.

My comments are the following:

  1. In this study, authors examined the CpG islands methylation status of a 13 genes set (ZAP70, ITGA4, KIF1A, PARP15, EPHX3, NTM, LRRTM1, FLI1, MIR193, LINC00599, MIR296, TERT, and GP1BB). Authors described that “previous studies, performed by our group, demonstrated the importance of epigenetic alterations and aberrant DNA methylation of specific genes to discriminate OSCC and its precursors lesions from benign oral mucosal lesions”. Please explain the detail of why authors picked up 13 genes.

Answer 1:

We modified the text accordingly as follows:

Section Introduction, page 2, line 73:

“Specifically, 19 genes known to be altered in OSCC [14,17-20] were evaluated by bisulfite Next Generation Sequencing (NGS) with the aim of developing a noninvasive procedure for OSCC detection based on oral brushing. ROC analysis of all CpGs investigated allowed us to select the highest informative ones mapped within the following 13 genes: ZAP70, ITGA4, KIF1A, PARP15, EPHX3, NTM, LRRTM1, FLI1, MIR193, LINC00599, MIR296, TERT, and GP1BB. A linear discriminant analysis was used to develop an algorithm of choice that clearly discriminated benign oral lesions from potentially malignant or malignant oral lesions [13,14].

None of our previous studies analyzed the impact of an altered methylation pattern of these genes obtained from pre-operative oral brushing samples on post-surgical clinical outcome. Recently, few studies from other authors demonstrated that genetic and epigenetic alterations of some genes of our 13-gene panel may play a role in OSCC prognosis, presence of metastasis and response to treatment [18,21-26]. For example, Marsit et al., starting from a case series of 68 post-surgical Formalin Fixed Paraffin Embedded (FFPE) OSCC samples revealed that an altered methylation pattern of ZAP70 and GP1BB is associated with poor survival [17]. Shintani et al., starting from 7 OSCC lines, showed that an altered methylation pattern of FLI1 is a prediction marker gene for OSCC radiotherapy resistance [22]”.

2. Authors examined the CpG islands methylation status in 36 OSCC cases. However, 75% (27 cases) are early stage (T1-T2). It is better to increase the number of advance cases (T3 & T4).

Answer 2:

We agree with the reviewer that the number of T3 &T4 OSCC cases is a limit of the present study. However, to overcome the limits of the small population study, we fit a Cox proportional hazards model via likelihood-based component-wise boosting, as reported in the statistical analysis section (page 11 line 354-382) and in the discussion section (page 7, lines 226-233). Boosting is a method for incrementally building linear combinations of “weak” models, to generate a “strong” predictive model and boosting algorithms have been shown to be particularly useful to handle models in which the number of candidate predictors exceeds the number of observations (high-dimensional settings). In the discussion section at page 9 line 290 we modified the following sentence in order to underline the limits of the present study related to T3-T4 cases.

“A limit of the present study is the low number of patients, especially the number of advanced OSCC tumors (T3-T4). However, component-wise boosting was specifically used to overcome the limits imposed by the small sample size. A second limit of the present study is the absence of a prospective test set of patients that we hope to undertake in future investigations in order to provide an unbiased predictive evaluation of the fitted model. A validated Cox model including molecular and clinical predictors has the potential to estimate for each patient the risk of disease relapse at any given time points following surgery.”

Furthermore, in our opinion identification of an altered methylation profile of 4 CpG sites of ITGA4 gene strongly related with prognosis in the homogeneous sub-group of 26 early stage OSCCs (T1-T2N0) represents the more interesting aspect of the present paper. Despite the apparent ease to manage these early staged tumors, sometimes the surgeon face to unpleasant surprises during the follow-up period. In fact, local recurrence can be experienced even if resection margins were clear and neck nodal metastases can appear in 10-30% of cN0 patients. Indeed, neck management for cT1-2 N0 patients remains debated. In these cases, known prognostic clinical-pathological factors (grading, depth of invasion, surgical margins, pattern of invasion) are usually less informative respect to advanced tumors. The presence of a biomarker with the ability of recognition of early tumors that will eventually show aggressive behaviors before surgery starting from pre-operative oral brushing specimen is of primary importance in oral cancer management. Some sentences to better highlight this aspect was added in the discussion section at page 8-9 lines 251-260.

“T1-T2 are commonly considered at lower risk of unfavorable outcome but they can pose a problem in the correct planning of surgical management. Despite the apparent ease to manage these early staged tumors, sometimes the surgeon faces to unpleasant surprises during the follow-up period. In fact, local recurrence can be experienced even if resection margins were clear and neck nodal metastases can appear in 10-30% of cN0 patients. Indeed, neck management for cT1-2 N0 patients remains debated. In these cases, known prognostic clinical-pathological factors (grading, depth of invasion, surgical margins, pattern of invasion) are usually less informative as compared to advanced tumors. The presence of a biomarker with the ability of recognition of early tumors that will eventually show aggressive behaviors before surgery starting from pre-operative oral brushing specimen is of primary importance in oral cancer management.”

3. Authors examined the methylation profile and clinical-pathological characteristics including tumor size and LN metastasis. How about other clinical-pathological characteristics? Ex) differentiation, invasion pattern, and/or recurrence…

Answer 3:

No differences were found in the methylation profile of 13 genes and other clinical-pathological characteristics of the studied population. A brief sentence was added in the results section at page 5 line 137-139.

“The Mann-Whitney U test and Kruskal-Wallis test did not show other differences in the methylation profile of 13 genes in relationship with other clinical-pathological characteristics of the studied population.”

Round 2

Reviewer 2 Report

I think that the authors satisfactory responded my comments. Therefore, I fell that this paper is acceptable for publication.